# Costa Resiliente: A Serious Game Co-Designed to Foster Resilience Thinking

Cristian Olivares-Rodríguez [1,*,†,‡], Paula Villagra [2,‡], Rodolfo E. Mardones [3,‡] and Luis Cárcamo-Ulloa [4,‡] and Nicolás Jaramillo [1]

1   Instituto de Informática, Facultad de Ciencias de la Ingeniería, Universidad Austral de Chile, Valdivia 5090000, Chile
2   Instituto de Ciencias Ambientales y Evolutivas, Facultad de Ciencias, Universidad Austral de Chile, Valdivia 5090000, Chile
3   Instituto de Estudios Psicológicos, Facultad de Medicina, Universidad Austral de Chile, Valdivia 5090000, Chile
4   Instituto de Comunicación Social, Facultad de Filosofía y Humanidades, Universidad Austral de Chile, Valdivia 5090000, Chile
*   Correspondence: cristian.olivares@uach.cl; Tel.: +56-940892499
†   Current address: General Lagos 2086, Valdivia 5111187, Chile.
‡   These authors contributed equally to this work.

**Abstract:** Resilience thinking is critical for improving disaster preparedness, response, and adaptation. While there are several strategies focused on assessing resilience capacity in human communities, there are few strategies focused on fostering resilience thinking. Game-based learning is an active and immersive teaching strategy that can foster complex skills such as resilience. However, this field needs further research in terms of its potential to strengthen community resilience to disasters. In this paper, we validated a serious game to foster community resilience. We present the collaborative creation process for the development of the board game Costa Resiliente, and its subsequent migration into a video game. We have developed an experimental study to evaluate the contributions of the mobile game against the board game. The result is a technological tool based on scientific knowledge to foster resilience thinking in coastal human communities exposed to hazards. The board game was developed using data from local research on community resilience, and from experts in emergency planning and developing games collected in focus groups. The board game's effect on fostering resilience thinking was validated with school students from a coastal town. During the migration process into a video game, we used a design thinking methodological approach for the co-creation of audiovisual elements, in which beneficiaries participate actively and early. Through this approach, visual and auditory elements that are familiar to coastal communities were integrated into the video game elements. Our study indicates that game-based learning is a useful approach to foster resilience thinking, and that a better gaming experience can be provided by a video game. The potential of this video game for educating young age groups about community resilience is further discussed.

**Keywords:** gamification; resilient thinking; co-design; coastal hazard; Chile

## 1. Introduction

Historically, the approach to risk management has been to identify hazardous zones in order to know where a disaster will occur and to generate mitigation strategies. Over time, it was necessary to assess the vulnerability of hazardous areas, since the magnitude of the disaster varies according to aspects of the territory such as the materiality of constructions and the socio-economic characteristics of communities (e.g., [1]). Knowing the vulnerability of communities allowed a change to the approach to disaster risk reduction (DRR) by developing preparedness strategies to reduce vulnerabilities. Following this shift, the multidimensionality of disasters became relevant, and the approach to DRR was complemented with recovery and reconstruction strategies. Today, most emergency agencies

worldwide rely on this traditional DRR approach to manage disasters, not recognizing that disasters should be managed differently according to every context [2]. The resilience approach evolved from the natural and psychological sciences to the urban and planning disciplines [3], as a contribution to DRR through the development of frameworks, checklists, tools, and resilience models that address the physical, social, environmental, economic, and institutional territorial dimensions, among others (e.g., [4–7]). Indeed, these ways of measuring and evaluating resilience can be modified and adapted to different contexts (e.g., [8]). This approach not only provides the knowledge to develop mitigation, preparedness, response, and recovery strategies, but also contributes with new knowledge to develop disaster adaptation strategies. These encourage managers and policy makers to increase the redundancy, diversity, and flexibility of territorial components, key for community resilience, and DRR [9].

Even though the resilience approach in disaster management literature dates to 1980 [10], it has been used more frequently in the last 20 years on a global level [11]. This is due to its importance as a factor in achieving sustainability [12], its role as a strategy for climate change adaptation [13], and as a requirement for communities to better respond and adapt to disasters such as 9/11 or Hurricane Katrina [14]. For the same reason, many non-governmental organizations, governments, communities, and civil society organizations are continually developing strategies to better prepare for hazards [15].

The resilience thinking approach [3] offers a particular way to understand and live in the context of disasters and to make communities more resilient. This approach seeks to reveal the significance of living in a cyclical and dynamic system, in which disturbances such as tsunamis, earthquakes, landslides, and fires are part of the landscape in which we inhabit. Coastal communities need to foster its resilience against natural disasters. There are incipient efforts where such methodologies to be aware about the territorial exposure [6] and practical tools to understand coastal disaster risk [7]. In particular, the Ministry of Education of Chile have defined proper learning goals in secondary school based on the biology of ecosystems and resilience [16]. Meanwhile, Chile's National Emergency Office (ONEMI) regulates the Integrated School Safety Plan (PISE). Such a plan is mandatory for every Chilean school by integrating the preparation for emergencies [17]. To think in a resilient way is to understand and to be aware of the cycles that humans experience, and to be better prepared and to adapt to unexpected and sometimes undesirable changes.

The current state of the environment, with large scenarios of climate shocks and socio-natural hazards, calls for a greater appropriation of skills and capacities in resilience by all society. Acquiring resilience thinking at an early age can lead to a resilient way of community life [18–20]. Learning at an early age has been shown to increase awareness about the sources of risks and the responsibility we have in our preparedness [21].

However, resilience is a complex skill that is hard to acquire through conventional methods such as lectures and workshops. New learning techniques such as gamification can provide solutions to this problem based on the emotional engagement produced by the challenges and rewards as a fast feedback to learners. In fact, there is robust evidence indicating that play has positive implications for cognitive, emotional, and physical development [22]. Therefore, using a game to foster resilience in children could catalyze not only their understanding, but also their projection in real disaster situations. This method could effectively foster resilience thinking from an early age and improve the community's ability to prepare, respond, and adapt to the effects of hazards.

### 1.1. Game-Based Learning

Active learning can achieve higher levels of learning retention than traditional learning sessions, with gamification being one of the most representative styles of active learning [23]. In a broader educational context, the problem of fostering and tracking certain social learning is being solved with gamification. Gamification comes from the idea of mixing game mechanics with any aspect of interactions, since the game has positive implications at the level of cognitive, emotional, and physical development [24]. In addition, games

awaken curiosity, are pleasurable, and help transmit essential skills to better cope with the world around us. Games teach aspects about how reality works, about understanding oneself, understanding the actions of others, and they stimulate the imagination, teaching users to minimize risks and to enhance decision making. Learning is considered to take place by practicing and making mistakes, from which limits and rules are tested [25].

In recent years, a trend has been to add game elements (ranking, scores, and leaderboards) to educational platforms [26]. Gamification improves the overall performance of a skill by adding metrics, and constant feedback can be given to the participant about their progress, triggering high levels of interest in learning [26]. Furthermore, play is attractive to different community groups, and even though play is usually understood as a practice that is associated with childhood, play has a social role in various age groups. For example, in children, it is associated with discovery motivations, and in adults, with cultural traditions or the care of cognitive conditions [27,28]. Currently, games are no longer only about individual and simple tasks to overcome stages and to obtain scores. Role-playing games articulated using tabletops and current networked games stimulate the development of strategic thinking, decision making, and collaboration in children, adolescents, and adults.

Today, the social interaction function of video games is a very interesting field to obtain more attention from young people and adults [29]. Meanwhile, data science and learning analytics methodologies can track player interactions in video games [30,31]. These techniques have been applied to museum contexts to track learning by capturing users' actions, decisions, and responses, before and during pedagogical visits led by a video game [32]. Thus, a video game can foster and massify the learning designed in a serious game, record users' actions, and expose a panel of analytics about the group participating in the training [33,34].

Among emerging taxonomies that promote certain aspects of education, serious games stand out. Serious games use elements of or a simulation of reality and mix didactic elements, to develop complex skills in a deeper way [35]. Serious games are defined as a mental challenge with an educational objective that serves to teach higher level skills such as inference and self-criticism [36]. By using elements of reality, one can better understand the consequences of actions without a real cost [37]. An indispensable feature of serious game design is the participation of subject matter experts who help to create the game [38]. In addition, concepts of fun and learning must be balanced in order not to affect the game's objective [39].

### 1.2. Active Learning and Serious Disaster Videogames

More precisely, active learning through games has been used to foster disaster education. Moradian and Nazdik (2019) analyzed the impact of disaster risk education by contrasting classical and other video game-based lectures in high school students [40]. In this scenario, participants in the video game group achieved significantly higher knowledge about disasters. Along the same lines, Delima and colleagues (2021) created DisCoord, a serious board game that proposes the participation of key community agents in the co-construction of knowledge about disaster management with a clear territorial focus [41]. This experience proved to be a pedagogical resource based on collaborative and active learning. Gampell and Gaillard (2016) analyzed the potential of video games in capacity building and disaster awareness, establishing a taxonomy based on prevention, mitigation, and preparedness strategies available in the analyzed games [42]. In particular, the inclusion of strategies such as the use of artificial structures for prevention, environmental policies for mitigation, and disaster risk analysis in preparedness were analyzed.

Solinska-Nowak and colleagues (2018) analyzed 45 serious games applied to disaster management education [15]. They observed that these digital learning environments provided a satisfying social experience for users during collaborative problem solving with an age- and knowledge-diverse audience. Most of the games analyzed were multiplayer, face-to-face with real-time interactions between players, and included hazards such as floods, volcanic eruptions, and tsunamis. In relation to the phases of disaster management, most

of the games included prevention and mitigation, and focused on knowledge transmission and an awareness of disaster management. In light of the above evidence about serious games applied to disaster education, the development of resilience thinking has received lesser attention.

The consideration of resilience thinking, with a focus on the adaptive capacity of communities in the face of disaster has been observed in two recent examples: the Multi-Hazard Tournament, in which a web-based decision support tool was used [43], and the Climate Adaptation Game [44]. Both seek to strengthen the adaptive capacities of communities when facing different hazards, which is emphasized in the game through collaboration among players and an increase in their knowledge of the territory, which are essential characteristics of a resilient community. In the Multi-Hazard Tournament, players must jointly evaluate alternatives that impact the resilience of the affected region, considering the game's objective, which is to maximize the benefits for the community with a limited budget. The participants in this game showed an increase in individual and community responsibility with respect to hazards. In the Climate Adaptation Game, players showed a high commitment to learning more about the benefits and needs arising from climate adaptation. The study of the Climate Adaptation Game suggests that it is essential to deepen the game experience, through improvements in game content, narrative, and others, to achieve better results.

Game experience is a relevant aspect to be considered from the game design to game evaluation. Game experience is defined through a psychological perspective, by a set of characteristics that are perceived by the individuals of gamification. In this sense, a positive game experience is based on a well-defined goal, clearly defined rules, and a proper statement of voluntary [45]. Thus, a game experience can be defined as a gameplay setting based on the sensations, feelings, actions, and meanings of players [46]. It can also be expressed as a co-creation between game and gamer [47], where players are immersed in scenarios by making decisions and meanings. Therefore, the game experience is a player's perception regarding player's perception of feelings, thoughts, and decisions, that influence the consideration of game repetition [48] and several assessments have been proposed [49,50].

## 2. Materials and Methods

We have considered the role of gamification, the value of serious games, and the scarce development of serious video games with a focus on resilience. The objective of this study was to develop a gamified technological tool that contributes to foster resilience thinking in human communities exposed to disaster. First, we have developed the Costa Resiliente board game based on a collaborative design approach. We have involved communities and experts in the design process. Secondly, we have validated the game potential to foster resilience thinking by school children between 10 and 18 years old from the Chilean coast of Southern Chile. In a third stage, we have migrated the game from a board version to a mobile version. Finally, we have evaluated the game experience for both versions of the game (Figure 1).

In this context we pose the following hypotheses:

1. The board game Costa Resiliente can develop capacities and abilities that foster resilience thinking in school children in Chile.
2. The game experience of the video game is better than that of the board game in terms of achievement, challenge, support, immersion, playability, and social experience.

These hypotheses were evaluated in two consecutive studies developed between 2019 and 2021. Both studies have a strong co-design component with key stakeholders of coastal communities.

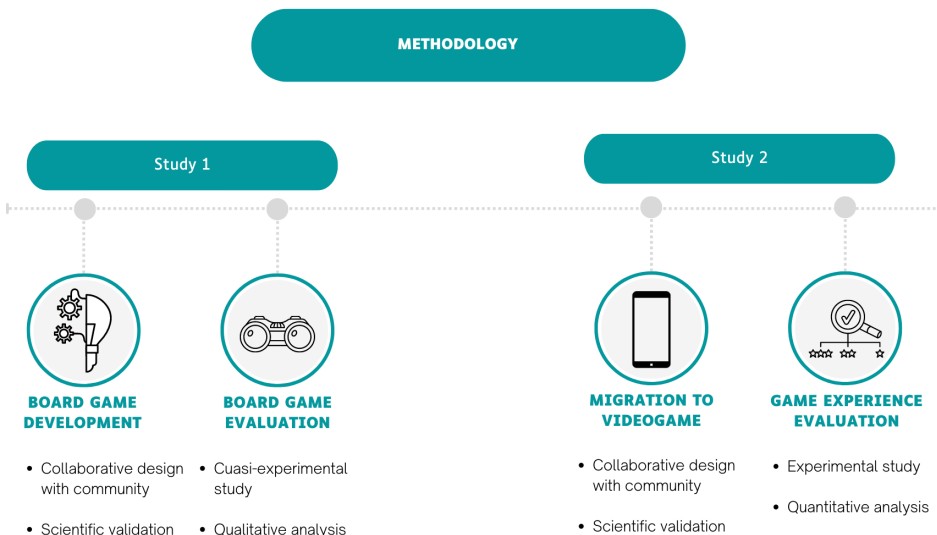

**Figure 1.** Study design.

## 3. Study 1: The Costa Resiliente Board Game

The Costa Resiliente board game was developed and evaluated in 2019. The initial idea was to create a game in which resilience strategies were learnt, to create more disaster-resilient communities through an adequate distribution of resources by key stakeholders in a coastal landscape.

### 3.1. Board Game Design and Development

3.1.1. Methodology

We have designed the rules, mechanics, challenges, rewards, and styles of the board game. Indeed, focus groups were used to reach a consensus on game elements, and to define the best mechanics for the gamification of these elements to foster a resilience thinking approach. Emergency experts from the Chilean National Emergency Office (ONEMI), resilience experts from the Landscape and Urban Resilience Laboratory of the Universidad Austral de Chile [51], and game experts from local companies (Diluviolúdico and La Tripulación SPA) participated in three focus groups. Input for these focus groups involved the experts' own experiences and community resilience research conducted on the Chilean coast [52,53], which was previously distributed to participants.

3.1.2. Results

Emergency and resilience experts identified and selected the following elements for the game: Six key actors, or roles for post-disaster community resilience on the Chilean coast (ONEMI, firefighters, policemen, the municipality, real estate agencies, fishermen, and ecologists); four hazards (tsunami, earthquake, landslide, and forest fires); 30 types of adaptive resources (e.g., wetlands, schools, and security areas); and 58 topics of general knowledge that are relevant for the Chilean coast context, which were later grouped into six themes.

In addition, resilience experts agreed that collaboration between actors is a key feature of a resilience community as well as noting that the distribution of resources should include strategies for their redundancy, diversity, and post-disaster survival (or robustness). Redundancy, or the repetition of a resource in the territory, is important for resilience, because if one resource is affected by a disaster, another can fill its role. Accessibility to diverse adaptive resources is also important, as there are multiple needs that arise after a disaster (e.g., access to water, shelter, and information). A resilience system should also be robust, meaning that the resources must withstand the effect of the disaster without receiving damage, degradation, or loss of function.

Considering this information, game experts suggested a role-playing board game with a focus on resource management and collaboration [51]. The game encouraged the idea of resilience thinking [3], since players should use redundancy, diversity, and survivability strategies to build a community that can be better prepared to cope with and adapt to hazards that can occur at any time or place.

The result was a game that includes two boards with different difficulties depending on the type of natural coastal landscape they depict, which vary in terms of the amount and distribution of beaches, dunes, wetlands, forests, and prairies. Each board is distributed in 36 quadrants that are transformed into neighborhoods of a town that should be resilient to different hazards. The game is played in rounds, with 3 to 5 players who take a role and distribute resources in each neighborhood. At the end of each round, players draw a query card; if the answer is right, they continue to the next round; but if the answer is wrong, they must draw a hazard card at random and follow the instructions about where the disaster occurs and its effects. Players are invited to act together and learn how to distribute resources to receive the least effect of the disaster while helping to reduce it. Hence, players collaboratively decide what is best for the creation of the town and its resiliency. At the end of the game, players count their points based on: Survival, or 1 point for each resource that remains on the board; Redundancy, or 1 point for each time a resource is repeated on the board; and Diversity, or 1 point for each time at least three different resources are found in a neighborhood.

### 3.2. Evaluation of the Board Game

### 3.2.1. Methodology

We conducted two 30-min game sessions with seventh- and eighth-grade students (N = 40) from the Rayen Lafquen School in the coastal town of Queule (Figure 2). Students were clustered in groups of five, and in addition, a team from the research groups acted as mediators. The mediator's role was to read the consultation cards and to ensure compliance with the rules. Game sessions were repeated after 3 weeks, and afterward, a guided discussion was conducted with the students. Both the sessions and group discussions were recorded and photographed prior to the request for informed consent.

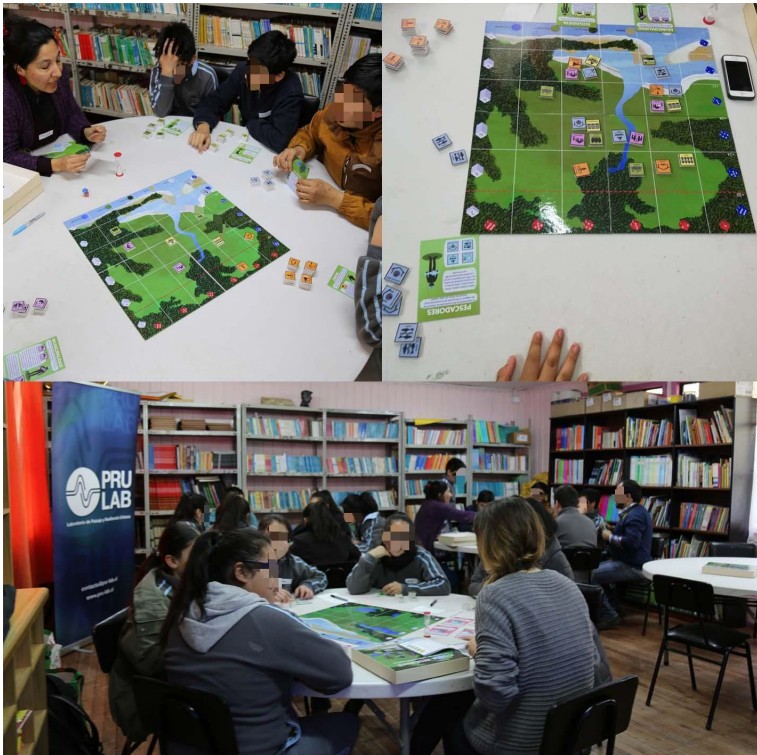

**Figure 2.** Playing Costa Resiliente at Rayen Lafquen School in Queule.

The analysis of this game experience was framed in a qualitative-inductive design, given its non-experimental nature, and sought to survey inductively the main ideas that emerged from the game, as stated by the participants themselves [54]. For this purpose, content analysis in Atlas-Ti software was conducted on the transcription of 10 audios, 4 of which corresponded to game sessions with eighth-grade students, 4 others to the seventh-grade students, and 2 corresponding to the conversations held with each grade separately. The content analysis included the reading and re-reading of the documents to delimit the most relevant themes and associations with the students' phrases, carrying out a thematic categorization that emerged from the dialogues [55].

3.2.2. Results

Ten skills and competencies emerged that can promote resilience thinking were identified after students played the game twice. First, concrete actions were taken by the students to achieve the redundancy (1) and diversity of resources (2). For example, redundancy is recognized when a student asks another student to place two equal resources in certain sites. One student mentioned *". . . I remember that it depends on the colors, and if there were different colors, it would give us more points"* (Boy, eighth grade).

Secondly, several emerging themes suggest the fostering of collaboration throughout the game, which is a resilient attribute of communities. Students showed empathy (3) by recognizing needs and actions in which another player is involved, e.g., *"I think it was a game of putting yourself in a person's place"* (Girl, eighth grade, group discussion). The fostering of collaboration is also observed in collaborative decision-making (4) when students put into practice the concepts of resilience and redundancy for distributing services on the board, i.e., *"Here place it next to it, or yes over there"* (Boy, eighth grade) or *"Surely a wetland over there? It could be right next to this."* (Boy, eighth grade).

Thirdly, the conversation during the game triggered procedural learning (5), as students made relevant decisions regarding how to strategically manage adaptive resources. For example, *"I would put the wetland here, here, or here because otherwise, the forest will burn"* (Boy, eighth grade). Or *"I would put a safe area where they can communicate"* (Boy, eighth grade). Conceptual learning (6) also emerged from data analysis suggesting an abstract knowledge about resilience, and the recognition of geographic components (7), by identifying natural elements that foster resilience. For example, *"Are the river and the sea good for the same thing?"* (Boy, eighth grade), *"What does wetland mean?"* (Boy, eighth grade).

Finally, other themes that trigger the game were detected, such as attitudinal strengthening (8) in the face of a disaster, linked to the capacity of empowerment observed in the students (9), and assuming roles during the game (10). This was observed in phrases such as *"I felt great as ONEMI, I looked like Super Man because I saved everyone"* (Boy, eighth-grade conversation).

In addition, the potential of the game in fostering community resilience was also observed in the final score of the students, which varied between 53 and 72 points, indicating *"good teamwork"*, which according to the game's scoring table, implied a good place to live, develop economically, and feel safe from disasters. Furthermore, the importance of mediation in the game was recognized in the intentional questions asked by participants before making decisions and/or verbalizing their interests, also promoting learning construction processes. For example: *"Where do we need more information points and shelters?"* (Mediator, eighth-grade session).

## 4. Migration to Videogame and Evaluation of the Mobile Game Costa Resiliente (Second Study)

### 4.1. Migration, Co-Design, and Development

We have move to a digital game in order to easily scale it and disseminate it to users, including its updates. The migration of Costa Resiliente from a board game to a video game was guided by a rigorous and systematic process that sought to explore emotions among participants to generate prototypes that were evaluated before being incorporated

into the software product. In particular, this project adopted Design Thinking [56], which provided a systematization of the process through the phases of (1) Empathy, where participants imagine the world from observation; (2) Definition, where participants agree on common aspects; (3) Generation of ideas, where they seek various alternative solutions and emerging constraints to the problem; (4) Prototyping, in which possible solutions are explored and visualized; and (5) Evaluation, in which all participants make judgments about the feasibility and viability of the prototype.

*4.2. Methodology*

In the design thinking model, value is provided by the diversity of members in the collaborative design process, providing validity and integration to the products being created [56]. For the development of the co-design, three 60 min collaborative work sessions were held, in which key project stakeholders participated, including the representatives of emergency management agencies, game development professionals, and representatives of the local community (ONEMI, La Tripulación SPA, the Municipality of Corral, and teachers and students from Corral schools). Under the Design Thinking model, the following steps were performed in each session: (1) Present the purpose and scope of the co-design session, (2) Organize the audience into teams, ensuring diversity, (3) Empathize with perceptions about the co-designed visual elements, (4) Define the visual elements, (5) Devise various design alternatives for the visual elements, (6) Prototype the visual elements, (7) Evaluate the prototypes, and (8) Close the session by thanking participants for their contributions and receiving final comments.

The information recorded in each session was captured in pre-designed worksheets, which allowed for the guidance and standardization of the co-design process among all working groups, as well as the persistence of the evidence. Subsequently, the templates were analyzed by the development team to evaluate their technical feasibility, relevance, and consistency, to subsequently design the prototypes that were integrated into the video game.

During the co-design process of the video game's audiovisual elements, prototypes developed in each of the sessions were evaluated. Specific questionnaires were developed for each type of audiovisual element that was submitted to co-creation with the project's key actors. These instruments were Likert-type surveys with a series of statements that had to be answered in the range of 1, strongly disagree; to 10, strongly agree. The following dimensions were considered:

- Identification and visualization: Does the co-designed prototype identify the real-life audiovisual element?
- Implementation and rules: Can the co-designed prototype be easily implemented, and follow the rules of the game?
- Identity and diversity: Does the co-designed prototype integrate identity and diversity of the real-life audiovisual element?

These dimensions of analysis were integrated into the evaluation of the Role and Hazard elements. Resources were analyzed according to their ability to identify the type of element, since the other dimensions had been evaluated by the development and research team.

4.2.1. Results

The migration of the Costa Resiliente board game to a mobile version was carried out through a co-design process of the audiovisual elements that had a territorial link with the beneficiaries. We have maintained the rules, mechanics, and challenges in both games. Several co-design sessions were undertaken to improve the styles of roles, hazards, and adaptive resources found in the board game. Tables 1–3 show the results indicating the co-designed definitions of roles, the effect of hazards, and the attributes of resources that emerged from the sessions. These definitions, effects, and attributes explicitly suggested a series of characteristics associated with each element, rooted in the experiences of partici-

pants in their territories, and were generally used to illustrate graphic pieces for the video game, as shown in the last columns of Tables 1–3.

**Table 1.** Co-design of roles.

| Role for Resilience | Co-Designed Definition | Element Attributes | Role Illustration |
|---|---|---|---|
| Fisherman: Ensures the local economy if a threat occurs, by installing infrastructures such as artisanal fairs. | A fisherman/woman is a person who embodies effort, is hardworking, perseverant, brave, and lover of work. | Yellow overall, boots, wool cap, an adult person. |  |
| ONEMI: Plans and coordinates resources to prevent and respond to emergencies (e.g., evacuation routes and safe zones). | Professionals from different areas whose purpose is prevention and/or control and mitigation of disasters for the benefit of the community. | Cargo pants, official ONEMI jacket, orange helmet, radio. |  |
| Real estate agencies build housing for permanent residences and for increasing tourism (e.g., houses and condominiums). | Empathetic, persuasive professionals, with extensive product and local knowledge. | Semi-formal, jeans, glasses, briefcase. |  |
| Ecologist: Decides where to create and grow ecosystems to mitigate and cope with catastrophes (e.g., wetlands and coastal forests). | A person that is passionate about caring for the environment, who gives much knowledge and practices to generate changes in the community. | Green T-shirt with leaves in the center, eco-style patches, waist bag, hiking boots. |  |
| Municipality: Builds and maintains public buildings that support the emergency (e.g., schools and hospitals). | Professionals from different areas, active and committed to social work and the community. | Semi-formal, denim, shirt. |  |

**Table 2.** Co-design of hazards.

| Hazard | Co-Designed Definition | Effects of Hazards | Hazard Effect Illustrations |
|---|---|---|---|
| Earthquake | Earthquakes generate a high impact on the community; it is important to know the affected environment and to have timely and accurate information. | Causes partial or complete destruction of structures (for the movable property), cracks in built spaces, the sinking of boats, and wetland drainage into water tables. |  |
| Tsunami | A catastrophe that causes a lot of damage at a general level (environmental, psychological, social, and economic). | Causes the disappearance of wooden buildings, with only the foundations remaining. Floors and buildings break in half, leaving debris and broken glass on the streets. |  |
| Landslide | A landslide has a strong impact on the earth, causing destruction with mud and earth movements, rumbling noises, affecting houses. | Causes ground noises, screams of people, and vehicle siren noises. Houses are filled with mud and often slide downslope. |  |
| Fire | These are uncontrolled fire events caused by natural and/or human actions, which generate material, human, economic, and environmental losses. | Causes post-event erosion (forests), total forest loss, a disparity effect/damage to ecological resources, and damage to emergency infrastructure. |  |

**Table 3.** Co-design of resources.

| Resilience Resources | Co-Designed Attributes | Resource Illustration |
|---|---|---|
| Fisherman | **Cove**: Moisture, Pier, Boats. **Boat rental**: Boats. **Restaurant**: Windows, White. **Craft Fair**: Premises, Food court, Open place. | |
| ONEMI | **Information points**: Map. **Shelters**: House, Wood. **Evacuation routes**: Signal, Route. **Safe areas**: Wide, Spacious, Grass. | |
| Real Estate | **Cabins**: Wood, Zinc roof, Wood heating. **Houses**: Trees, Plants (Vegetable garden). **Apartments**: Cement, not so high (2–3 floors). **Luxury housing**: High, Rustic facade, Spacious. |  |
| Ecologist | **Forest**: Diversity of native species. **Dunes with vegetation**: Sand, Dune relief, Specific vegetation type. **Wetland**: Flora, Water, Birds, Mist. **Prairie**: Large extensions of pasture, without noticeable reliefs, livestock. | |
| Municipality | **Firemen/women station**: Fire trucks, the predominant color red, siren. **Police station**: Concrete buildings, green and white, patrols outside the building. **School**: Concrete buildings, Bus. **Hospital**: Interior parking, Cream color, Ambulance, Concrete building. | |

The illustration of roles, hazard effects, and resources or prototypes, were positively evaluated by the participants in relation to the three dimensions of evaluation: identification, implementation issues, and identity/diversity. Co-design participants clearly recognized the roles as community members (identification), fit the game mechanics developed in the first study (implementation issues), and met diversity and identity present in the real world (diversity). Therefore, each participant evaluated every co-designed element during the sessions as being high or strongly high.

*4.3. Evaluation of the Gaming Experience*

The Costa Resiliente video game developed based on the products described in Section 4.2.1, was evaluated against the board game version through a quasi-experimental design. This evaluation was performed with children and adolescents from the coastal city of Corral, in Southern Chile. Participants accepted the informed consent of the study. Two study groups were randomly formed: Board game (N = 17) and Video game (N = 34), who autonomously used the corresponding version of the game for two months.

### 4.3.1. Methodology

A questionnaire was applied after participants played the games to determine the differences in game experience between the board and mobile game versions. The GAME-FULQUEST instrument was used because it has been shown to determine the experience of using a video game through seven dimensions of analysis [50]. The instrument was a questionnaire with 64 statements distributed among each dimension. Statements were answered on a Likert scale between 1, strongly disagree and 5, strongly agree. The dimensions were defined by the authors as follows:

- Achievement: Experiencing the demand or drive for success in performance, goal attainment, and progress.
- Challenge: Experiencing where the person's ability is put to the test.
- Competition: Experiencing rivalry towards one or more actors to obtain a scarce and desirable outcome for all actors.
- Support: Experiencing guidance on how, what, and when to do something, and how to improve the target behavior.
- Immersion: Whole attention is captured, and the person is absorbed in what he/she is doing while having the feeling of being dissociated from the real world.
- Social experience: Experiences that emanate from the direct or indirect presence of people, the social actors created by the service, and the service as a social actor.
- Playability: The experience of engaging in voluntary and pleasurable behaviors driven by imagination or exploration, while being free or under spontaneously created rules.

The Competence dimension was discarded from the analysis, as it is not relevant for Costa Resiliente, which is a collaborative role-playing game, in which participants must make decisions together.

### 4.3.2. Results

In the Achievement dimension, the participants who used the video game positively evaluated the game features, independent of its modality. However, it was in the video game where it was observed that participants felt a higher level of challenges posed in the game mechanics. In the Challenge dimension, participants evaluated the game positively, regardless of the modality used. However, the participants who used the video game had a better perception of the level of challenge involved in the gameplay than those who used the board game. This may be due to the automation of the game rules, which contributes to a greater degree of concentration on the activities proposed in the design of the mechanics. In the Immersion dimension, a greater difference in perception was observed between those who used the video game versus the board game. The former achieved a better experience, mainly because the game rules are automated, and attractive audiovisual elements are integrated for the community. In the Playability dimension, participants reported having a positive perception regarding the dynamics and activities proposed by the game. Likewise, a small difference was appreciated between both game modalities, which confirms the appropriate design of the mechanics and elements of the game, as they transcended the environment. In the social experience dimension, participants showed some of the highest evaluations among the dimensions of the questionnaire, regardless of the modality used. However, a higher positive tendency was observed among those who used the video game, which may be related to the automation of the rules, which reduces the cognitive load of the game and encourages the emergence of emotions derived from the ludic proposal [57]. Finally, in the Support dimension, participants declared to have a positive perception regarding the support elements in the understanding of the rules and dynamics of the game, independent of the modality used. However, the video game provided a better perception than the board game, which can be linked to the reduction in cognitive load by means of relevant and active messages that both warn and help users.

Figure 3 shows the responses (vertical axis) of the participants for each of the six dimensions of game experience (horizontal axis). In all the dimensions of analysis, participants

reported a positive perception of their gaming experience, which was quite similar between game versions. However, the game experience was slightly improved when using the mobile version, since in all dimensions the mean was higher and the dispersion was lower compared to those who used the board game version. In short, the average evaluation slightly favors the video game version in all dimensions, and the widest score variations are observed in immersion and challenge for the board game, and in support and social experience for the video game version.

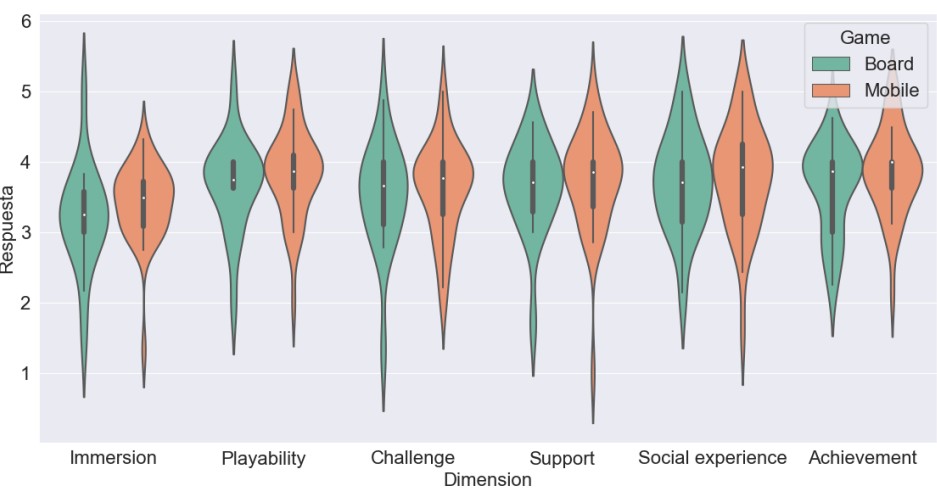

**Figure 3.** Results of the questionnaire for the game version: board and mobile.

## 5. Discussion

An evaluation of the results from the Costa Resiliente board game confirmed the game's potential to foster resilience thinking in school children, confirming our first hypothesis. Costa Resiliente encourages collaboration among players by taking on different roles. In addition, the game helps to develop abilities and knowledge to use resilience strategies, such as redundancy and diversity, in resources distribution.

From the point of view of resiliency, these results indicate that the game can develop social capital, which is a relevant characteristic of resilient communities, as it affects the capacity for organization and innovation after a disaster to respond positively together [3,58,59]. In addition, the game contributes to developing players' resilient-related knowledge, whether abstract or concrete, when, for example, players consult each other about the meaning of the elements that they have on the board (e.g., wetlands). The role of objective knowledge, e.g., how much people really know about a hazard and how to deal with it, is also a characteristic of a resilient community because it is needed for resilient behavior. For the Chilean context, this is an interesting finding because objective knowledge has been incorporated into school programs by means of lectures and workshops in classrooms organized by local emergency offices. In addition, educational plans and programs have incorporated the objective of explaining the effects of climate change on biodiversity, biological productivity, and ecosystem resilience, as well as its consequences on natural resources, people, and sustainable development [16]. This concern is present in a small part of the science elective curriculum for the final years of secondary school. Costa Resiliente can contribute to increasing objective knowledge in a playful way, facilitating learning. This is a critical point because the way in which resiliency is being framed and communicated to the community is crucial to fostering inhabitants' preparedness and response to disasters [60]. Costa Resiliente is intended as a serious game, and in its design and development process, it has been conceived as an educational input for both school and family contexts.

From the point of view of the contribution of gamification to foster resilience thinking, our results indicate this is a fruitful approach. Active learning, mainly game-based, has

shown to yield higher levels of learning retention than traditional learning sessions [23], higher engagement with learning regarding climate adaptation by participants [44], and greater appropriation of knowledge regarding disasters, compared to using classical strategies [40]. Thus, we developed a learning game through a collaborative process that integrated communities into its design, but also constrained the product in game-based education to foster the retention and engagement of players. Therefore, we developed an active learning resource for sustainable development through a collaborative role-based serious game that integrates the process of urban planning and resource management in a playful way.

Consequently, game-based active learning has shown evidence for participants' interest in disaster management when incorporating real-world elements present in the territory. Video games have provided greater interaction, dynamism, and feedback compared to other types of games, contributing to a better gaming experience. As such, Costa Resiliente showed a positive gaming experience among players, and the video game improved gameplay compared to the tabletop version, confirming our second hypothesis.

This work has presented the definition of clear mechanisms of deliberative participation of the community in the co-design and co-creation of audiovisual elements that are part of Costa Resiliente in its mobile version. Each participant had an opportunity to contribute with elements from their environment, participate in the design, and be a judge in the evaluation of the co-created prototypes. This contributed to the appropriation and relevance of the developed educational resource, as observed in the literature [41,61–65]. The co-design of the audiovisual pieces incorporated in the video game demonstrated a strong commitment by the relevant agents of the community, which is appreciated in each of the stages of the process. It is observed that the participants were actively involved and participated both in generating the characteristic elements of the co-designed pieces and in elaborating the digital prototypes. This is observed in the definitions co-created among the participants, who reflected their territorial experiences in the construction of the prototypes, which led to the construction of the final designs. Likewise, participants positively evaluated the quality and relevance of the designed prototypes. Therefore, our results are in line with previous experiences of involving the community from the early stages of the development process, influencing user commitment and engagement.

Results from the game experience show coherence with those reviewed in the literature regarding the commitment and involvement of trainees during active learning sessions. However, our proposal is an advance in terms of the evaluation of the game experience by using two modalities of the same version of the game. In both games, the rules and mechanics were maintained, and only the medium was modified. This is the first time in the literature that the impact on the playability of two modalities of the same serious game is evaluated. Among the main results of the second study, we observed a similar gameplay experience between both modes, physical and digital, but we tended to improve among those who used the video game. The migration of the board game to a digital version through a mobile application contributed to participants' gaming experiences, since it improved the perception of playability, immersion, social experience, and support. These results are supported by the automation of rules and mechanics through algorithmic means, which reduces the cognitive load of the game and allows participants to focus their actions on the decisions, and the discussions designed in the game. The improvement in the social experience dimension is noteworthy, which is associated with the improvement in the relationship between participants due to the elimination of a dependence on a monitor to regulate the development of the game. However, it is necessary to advance via the integration of new functionalities that contribute to the game experience over time, moving from sessions focused on games to sessions that evolve game by game, and team by team, improving the adherence of the players. Likewise, it is necessary to establish game models that allow flexibility in the participation of different types of players, and even non-players, in the development of resilience skills. Finally, it is necessary to incorporate the components

of social interaction between teams to contribute to the formation of a resilient community around the technologies provided by Costa Resiliente.

Consequently, the Costa Resiliente Serious Game, in both modalities, was positively evaluated as a satisfactory social experience. This experience is key for the development of knowledge and skills through active learning based on games, delivering instances of real-time interaction among the participants of each game, and being aligned with hazards of greater incidence in the territory [15]. Likewise, the developed game provided timely feedback and clear support to players regarding the outcome of their actions and the achieved learning, which is essential for players to have satisfactory progress and correct learning. These results were consistent with recommendations in the related literature [42]. Therefore, this serious game has the necessary conditions reviewed in the literature for users to improve their skills.

## 6. Conclusions and Future Work

The evaluation of the collaborative role-based serious game, Costa Resiliente, has been shown to develop resilience thinking among school students through an active learning strategy. In addition, the mobile version was tested to provide a better game experience because participants' perceptions were more positive regarding each evaluated dimension against the board version. More precisely, users felt more supported, immersed, and socially disposed of, because the mobile game encapsulated the rules and provided help to foster the player's communication during each game session, which was not possible in a board game session.

The Costa Resiliente video game is based on scientific knowledge for game mechanics design, based on co-design processes during the creation of game elements, and based on the rigorous evaluation of artifacts. In addition, the video game architecture was developed to track community resilience through the recording of participant actions to know their planning decisions, game by game. In this sense, the video game was able to identify the objective knowledge regarding resilience based on users' answers to the questions embedded in the video game. Such functions provided a set of user behavioral data that allowed the computation and validation of learning analytics derived from the game, and they are linked to the resilience thinking of the users. Therefore, we contributed an innovative technology that is able to foster resilience thinking in communities exposed to disasters. We achieved this through active learning and the generation of key information for decision-makers in disaster management.

Knowing that the mobile version had a greater capacity to foster resilience thinking opens future possibilities to increase resiliency due to the technological characteristics of this version of the game. In the mobile version, it is is easier to provide scalability and updates of the game to add new knowledge, resources, roles, and landscape types, among other strategies that can make the game more interesting and challenging in the future, assuring its sustainability over time. In addition, the mobile version has more potential for massification than the board game, and could reach other schools in Chile and in other countries. More importantly, the technology of the mobile version, including all its attributes and potential for scalability and massification, can be complemented with an analytic board in which players' behaviors in the game could be collected and evaluated online. Therefore, it can be possible to know how the resilience thinking of the community changes over time, positively or otherwise. Moreover, the use and access to this technology by teachers, schools, and other type of education programs for children can be of great value in improving and diversifying education programs about disaster. For instance, in Chile, Costa Resiliente can be included in current programs developed by the Ministry of Education, the ONEMI, and the Institute for Disaster Resilience, aimed at teaching children about risk and how to be more prepared for disasters. The use of the Costa Resiliente by these organizations can contribute to building community resilience from an early age.

**Author Contributions:** Conceptualization, C.O.-R. and P.V.; methodology, C.O.-R. and P.V.; software, C.O.-R. and N.J.; validation, C.O.-R. and P.V.; formal analysis, C.O.-R. and P.V.; investigation, C.O.-R., P.V., R.E.M., and L.C.-U.; resources, C.O.-R. and P.V.; data curation, C.O.-R., P.V., R.E.M., and L.C.-U.; writing—original draft preparation, C.O.-R.; writing—review and editing, C.O.-R., P.V., R.E.M., and L.C.-U.; visualization, C.O.-R.; project administration, P.V.; funding acquisition, P.V. All authors have read and agreed to the published version of the manuscript.

**Funding:** This research was funded by FONDEF IDeA I+D, grant number ID20I10091, titled "Desarrollo, Investigacion y Validación de un Videojuego para el Fomento de Aprendizajes Sociales Frente a Desastres de Origen Natural en Comunidades Costeras".

**Institutional Review Board Statement:** The study was conducted in accordance with the Declaration of Helsinki, and approved by the Institutional Review Board (or Ethics Committee) of Universidad Austral de Chile (Res. N° 23/214 and 2 December 2020).

**Informed Consent Statement:** Informed consent was obtained from all subjects involved in the study.

**Data Availability Statement:** Not applicable.

**Acknowledgments:** Acknowledgements are made to the communities of Corral, La Aguada, San Carlos, Huiro, Chaihuin, San Juan, and the surrounding areas that voluntarily participated in this process. Finally, we would like to thank the Municipality of Corral and ONEMI of the Los Ríos Region.

**Conflicts of Interest:** The authors declare no conflicts of interest.

## Abbreviation

The following abbreviation is used in this manuscript:

ONEMI    Chilean National Office for Emergency

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
