# Peer review of "Costa Resiliente: A Serious Game Co-Designed to Foster Resilience Thinking"

_sustainability, doi:10.3390/su142416760_

Round 1

Reviewer 1 Report

You, the authors, did a great job of clearly and concisely demonstrating the benefits of gamification as well as mobile games in a scholary context. The topic is interesting and important for future education. I especially think the concept can be utilised in other areas (eg here in Austria) for a different set of emergencies and I am looking forward to reading more of your research.

Author Response

Dear reviewer,

Thank you very much for your comments. We are interested in to scale the game by incorporating new features and making other validations in other contexts.

Best regards.

Reviewer 2 Report

Note to everyone:

When I do a review, I comment on the paper as I read it. I realize readers will tend to jump around, but if we writing something as a narrative, I will read it that way. So ... my comments will often seem like a stream of consciousness. After the fact, I'll note "main issues," which you'll see above the reviews I write as I read. In some cases, I may even contradict myself, which means somehow I missed something in the flow, which indicates a possible area to clarify. 

Main issues to address:

1. See Line 149. I think the paper limits the hypothesis to regional children, but the way it reads implies ALL children. The study improves the case for games+resilient thinking, but it can't prove such a sweeping implication based on the sample size. I suspect it's a wording issue, but the authors need to clarify the issue. 

2. I think the grammar has issues that purchasing a grammar-checker/adviser could resolve. I attracted a screenshot from Grammarly--but I also realize that it reports a lot of false positives. So when you see the blizzard of squiggles, please keep that issue in mind.

3. I'm finding that my comments nitpick, for lack of a better term. From my perspective, I consider it a sign of a good paper--I'm trying to fix smaller items. It also means everything comes across as negative--I apologize in advance. I blame academia, which is an issue far beyond this review.

4. The comparison between the board game and videogame doesn't really tell us anything. See also line 428 and Section 6. The authors make a stronger case for a digital version, but I'm still unconvinced. I recommend thinking about the case for a digital version, who's going to keep maintaining, why it more readily handles changes, and how it can disseminate more than the paper version. For example, printing a paper game and looking at rules online may slow down gameplay, but teachers and others could readily print everything and customize immediately. Digital games take a lot more consistent effort.

****************

Out of curiosity, I loaded the paper (minus images and references to reduce file size) into Grammarly. I took a couple of screenshots and loaded them into this review. I realize it costs more for everyone, but the authors should probably spend the funds and get a license to fix and adjust the wording in the paper. I've given up trying to edit my writing at this point--you just can't find everything. Note that my screenshot shows a variety of false positives (like line numbering, attempting to translate from other languages, etc).

Abstract:

"scarcely proven"--there are preliminary results. I know, because I've conducted some of this work. Limited results? Preliminary? Maybe better to say this field needs further research. 

Section 1. Introduction (after this section, I'll just use section numbers)

(around line) 40: should spell out why games and play are effective -- they can engage more than a lecture 

1.1:

52: "with any space of social life" -- unclear -- maybe you mean "any aspect of interactions?"

1.2

126: maybe it's a matter of style -- I like when authors italicize terms, especially if they'll be used throughout the paper -- see "Game experience" -- I realize I didn't flag the idea earlier. Unsure if MDPI wants that style.

134: Maybe a matter of style guide again -- "his/her" is out -- "their" is in -- or consider rewording as "a player's perception of feelings, ..." -- you don't need the pronouns. "his/her" appears elsewhere in the paper. Best just to remove it.

Section 2:

138: style issues -- keep sentences about 1-2 lines each -- see if you can make sentences active. "The authors sought to ..." or even just "We set the goals as ..." -- see what MDPI suggests. "I want to replace passive writing when possible" vs "I'm seeking to eliminate passive writing." (interestingly, sometimes passive is shorter :-) You can't always find a clear version, but I recommend trying. 

142-143: an example of why passive fails -- who developed the game? the authors? student researchers?

143: maybe I missed something -- what is "co-design?" Could be me: https://www.beyondstickynotes.com/what-is-codesign. Maybe give one reference.

148: a case for Grammarly or similar: "We put forward" becomes "We pose"

149: the board game fosters in ALL school children? Or just those in the region? See 153-155 -- maybe define the communities as target of study first? Did I understand that correctly?

Section 3:

3.1.1: answers what I asked earlier. Maybe just say earlier that Section 3.1.1 provides details on a multidisciplinary group of designers and developers as stakeholders ... something like that. A bit jargony, but lets authors answer my issue early without spending too much time up front. Also points out early that it's not just "a bunch of random faculty" inventing something out of thin air :-)

188: Now I'm wondering what the role of the authors was. Did you facilitate the focus groups?

193: I recommend a separate subsection on just "Game Design" -- explain what got created -- when I get to Section 4, I ask myself, "What exactly is the board game? the rules? Can I access it?"

Section 4:

265: why did the researchers decide on a digital game? Why not keep working with the boardgame? See Section 4.3.2 -- I'm trying to understand if this other section is answering that question.

318 and Figure 3: I think the authors need to clarify the comparison. If the videogame exactly copies the boardgame, then the comparious might be interesting. But if they compare two different games, the results don't really tell us anything. Say a user struggles a digital interface--is the game worse in terms of gamifying resilience? Not necessarily. 

Section 5:

428: I'm not convinced. Table games can be quite compelling. Are the authors making this claim for their videogame?

Section 6:

Call this section "Conclusions and Future Work"

503: This paragraph seems to be answering my question in 428. However, I'm still not convinced. Who will continue to update and fix the code? Paper versions can be printed and modified. 

Author Response

Dear reviewer,

Thank you very much for your comments, we have integrated them into our article according to the line of argument.

Best regards.

Reviewer 3 Report

Aim of the work is to present the process of co-creation for the development and validation of the board game Costa Resiliente, and the subsequent migration into a video game. Some suggestions are provided in order to improve the work:

Abstract

Clearly state the aim of the work and the methodological approach used in a better way

Introduction

The concept of resilience in disaster management literature should be analized more in depth, by also saying what are the main field of analysis of the topic that exist in the literature. Some useful reference on this could be: Anelli, D., Tajani, F., & Ranieri, R. (2022). Urban resilience against natural disasters: Mapping the risk with an innovative indicators-based assessment approach. Journal of Cleaner Production, 371, 133496 and Van Dongeren, A., Ciavola, P., Viavattene, C., De Kleermaeker, S., Martinez, G., Ferreira, O., ... & McCall, R. (2014). RISC-KIT: resilience-increasing strategies for coasts-toolKIT. Journal of Coastal Research, (70 (10070)), 366-371.

Moreover, could be important to say something also on the actual policies and tools that exist for improving resilience knowledge on the geographical context considered.

2. Materials and Methods

The figure 1 isn't clear, please revise it in order to be more simple and clearly states the process of the methodology

Author Response

(The authors gave the same response as above.)

Round 2

Reviewer 3 Report

The efforts made by the Authors are apprecciated. A major improvement of the introduction section regarding the disaster risk management could be suggested.

Author Response

Dear reviewer,

Thank you very much for your comments, we have improved the introduction.

Regards.
